# Blastocyst complementation using *Prdm14*-deficient rats enables efficient germline transmission and generation of functional mouse spermatids in rats

Toshihiro Kobayashi [1,2,10 ✉], Teppei Goto[1,10], Mami Oikawa[1,10], Makoto Sanbo [1], Fumika Yoshida[1], Reiko Terada[1], Naoko Niizeki[1], Naoyo Kajitani[3], Kanako Kazuki[3], Yasuhiro Kazuki[3,4], Shinichi Hochi [5], Hiromitsu Nakauchi [6,7], M. Azim Surani [8,9] & Masumi Hirabayashi [1,2 ✉]

Murine animal models from genetically modified pluripotent stem cells (PSCs) are essential for functional genomics and biomedical research, which require germline transmission for the establishment of colonies. However, the quality of PSCs, and donor-host cell competition in chimeras often present strong barriers for germline transmission. Here, we report efficient germline transmission of recalcitrant PSCs via blastocyst complementation, a method to compensate for missing tissues or organs in genetically modified animals via blastocyst injection of PSCs. We show that blastocysts from germline-deficient *Prdm14* knockout rats provide a niche for the development of gametes originating entirely from the donor PSCs without any detriment to somatic development. We demonstrate the potential of this approach by creating PSC-derived *Pax2/Pax8* double mutant anephric rats, and rescuing germline transmission of a PSC carrying a mouse artificial chromosome. Furthermore, we generate mouse PSC-derived functional spermatids in rats, which provides a proof-of-principle for the generation of xenogenic gametes in vivo. We believe this approach will become a useful system for generating PSC-derived germ cells in the future.

[1] Center for Genetic Analysis of Behavior, National Institute for Physiological Sciences, Okazaki, Aichi, Japan. [2] The Graduate University of Advanced Studies, Okazaki, Aichi, Japan. [3] Chromosome Engineering Research Center, Tottori University, Yonago, Tottori, Japan. [4] Division of Genome and Cellular Functions, Department of Molecular and Cellular Biology, School of Life Science, Faculty of Medicine, Tottori University, Yonago, Tottori, Japan. [5] Faculty of Textile Science and Technology, Shinshu University, Ueda, Nagano, Japan. [6] Division of Stem Cell Therapy, Distinguished Professor Unit, Institute of Medical Science, The University of Tokyo, Minato-ku, Tokyo, Japan. [7] Institute for Stem Cell Biology and Regenerative Medicine, Department of Genetics, Stanford University School of Medicine, Stanford, CA, USA. [8] Wellcome Trust/Cancer Research UK Gurdon Institute, University of Cambridge, Cambridge, UK. [9] Department of Physiology, Development and Neuroscience, University of Cambridge, Cambridge, UK. [10] These authors contributed equally: Toshihiro Kobayashi, Teppei Goto, Mami Oikawa. ✉email: tkoba@nips.ac.jp; mhirarin@nips.ac.jp

Genetically engineered animals provide essential insights into in vivo gene function and human genetic disorders. Pluripotent stem cells (PSCs) have been routinely used for targeted gene manipulation, particularly in rodents. PSCs can form all the tissues of the body, including the germline. Upon injection into the morula/blastocyst stage embryos, PSCs can carry their genetic modification to the next generation via germline transmission. While recent progress in genome editing technologies such as CRISPR/Cas9 in zygotes have enabled the generation of desired animals rapidly and efficiently[1,2], PSCs are useful for precise genetic modifications[3]. Introduction of point mutations or genes of interest via gene targeting and complex cell-based genetic modifications, such as microcell-mediated chromosome transfer, rely on PSCs[4,5]. However, germline transmission of the genetic modifications by chimerism with PSCs can be variable and inefficient, partly because of the competition between the host and donor cells in chimeras that are affected by the quality of PSCs. An experimental approach that results in predictable functional gametes with desired genotypes from PSCs more efficiently is highly desirable. Here we apply the blastocyst complementation method to overcome the variable germline transmission of genomic modifications.

Blastocyst complementation can be used to generate a tissue or organ from donor PSCs in genetically modified animals. For example, injection of PSCs into blastocysts derived from animals that are unable to make certain tissues or organs allows donor PSCs to develop and compensates for the missing tissues or organs in chimeric animals. So far, other groups and we, have demonstrated the successful generation of several organs, such as the pancreas, thymus, kidney, liver, brain, and lung by blastocyst complementation[6–12]. Such an approach is also possible in an interspecific setting between the mouse and rat[6,10,12]. If the same strategy can be applied to the germline using germline-deficient animals, PSCs could develop efficiently as gametes even when the quality of PSCs is not optimum.

For a suitable host blastocyst lacking the germline, we focused on *Prdm14*, a key transcriptional regulator essential for the specification of primordial germ cells (PGCs), the founder cells for gametes. In rodents, a lack of *Prdm14* causes loss of PGCs during development, which subsequently leads to infertility both in males and females. Importantly, the development of other organs and tissues is not impaired in these mutants[13,14]. Therefore, *Prdm14*-deficient animal could serve as an appropriate host for the derivation of germ cells from donor PSCs.

Here, we demonstrate efficient germline transmission of PSCs via blastocyst complementation using germline-deficient *Prdm14* knockout (KO) rats[14]. The rat is an important organism for physiological and pharmacological modeling of human diseases[15]. Achieving appropriate culture conditions to maintain germline-competent rat embryonic stem cells (ESCs) has enabled generation of genetically modified rat.[16–19]. However, rat PSCs are less robust than mouse PSCs, partly because of the difficulties of handling and clonal selection after genetic manipulation[20,21]. Besides, rat PSCs exhibit rapid loss of genomic imprints in culture, which affects their developmental potential, as judged by tetraploid complementation experiments designed to produce an entirely PSC-derived animal[22,23]. Here, we show that the use of rat blastocyst from *Prdm14* KO rats serve as excellent hosts for the germline transmission of donor cells. We use our method to demonstrate the production of a double mutant rat line, and rescue of germline transmission of PSCs with a mouse artificial chromosome (MAC). Finally, we also show the derivation of viable mouse gametes using the *Prdm14* KO rat host blastocysts.

## Result

**Blastocyst complementation using *Prdm14* KO rats enables efficient germline transmission of allogenic embryonic stem cells (ESCs).** First, we tested if host *Prdm14* KO rat embryos lacking germ cells permit the efficient contribution of exogenous rat PSCs to the germline. For this, we collected *Prdm14* KO rat blastocysts, by crossing *Prdm14*^HV/+ with *Prdm14*^mut/+ rats (both with albino coat color: Fig. 1a). We selectively used Prdm14-H2BVenus positive blastocysts (*Prdm14*^HV/+ or *Prdm14*^mut/HV) to track *Prdm14*, since it is highly expressed in the inner cell mass of the rat blastocysts[14]. For donor rat PSCs, we used male and female rat ESCs (black coat color) ubiquitously expressing tdTomato established from homozygous *Rosa26-tdTomato* knock-in rats[24] (see "Methods"). After injecting ESCs into Prdm14-H2BVenus positive blastocysts, we transferred them into the uteri of foster mothers and analyzed their contribution towards the germline in the adults (Fig. 1b). In the *Prdm14* heterozygous host (*Prdm14*^HV/+) where germ cells develop normally akin to wildtypes, rat ESCs contributed to somatic lineages and the germline depending on the extent of chimerism (Fig. 1c, d, Supplementary Table 1). After crossing with wild-type rats, a small proportion of offspring showed tdTomato fluorescence and dark skin (male: 16/174 [9.2%], female: 10/87 [11%], Fig. 1g, Supplementary Fig. 1a, Table 1, Supplementary Table 1), suggesting that germline transmission of donor rat ESCs occurred only in some neonatal rats. In contrast, in the *Prdm14* KO host (*Prdm14*^mut/HV), rat ESCs contributed to somatic lineages like in the *Prdm14* heterozygous rat, but all the germ cells in sex-matched chimeras showed tdTomato fluorescence (Fig. 1e, f, Supplementary Table 1). Whereas, the *Prdm14* KO host has impaired PGC specification and development, and does not show any detectable germline contribution. Notably, all the next generation rats obtained after crossing with wild-type rats exhibited tdTomato fluorescence and dark skin (male: 85/85 [100%], female (31/31 [100%], Fig. 1h, Supplementary Fig. 1a, b, Table 1, Supplementary Table 1), suggesting complete germline transmission of donor rat ESCs without any host-derived germ cells. Collectively, we find that blastocyst complementation using host *Prdm14* KO rat blastocysts enabled efficient germline transmission of donor rat ESCs.

**Efficient production of genetically modified rats using blastocyst complementation in the germline.** Next, we investigated whether blastocyst complementation in the germline is a viable tool to generate genetically modified rats efficiently. To demonstrate this, we used *Pax2/Pax8* double mutant rat ESCs as donor cells to obtain double KO rats. Both *Pax2* and *Pax8* are transcriptional regulators important for the proliferation and differentiation of multiple cell types, particularly in the kidney[25,26]. Importantly, the *Pax2/Pax8* double KO mice show a complete lack of kidneys as well as ureters. *Sall1* KO animals are the closest anephric model for the *Pax2/Pax8* double KO mice, lacking renal parenchyma; however, they retain ureters[7,10]. Generating double KO's using the conventional method, by crossing heterozygous mutants is highly inefficient with KO's expected at a 1/16 Mendelian ratio. Thus, here we applied blastocyst complementation in the germline for efficient generation of *Pax2/Pax8* double mutant rats.

We first screened male and female *Rosa26-tdTomato* rat ESC lines that have a frameshift mutation in *Pax2* and *Pax8* coding sequences created with the CRISPR/Cas9 system (Supplementary Fig. 2). Then, we injected them into *Prdm14*^HV/+ or *Prdm14*^mut/HV blastocysts (Fig. 2a). While donor ESCs lacking *Pax2/Pax8*, are expected not to form the

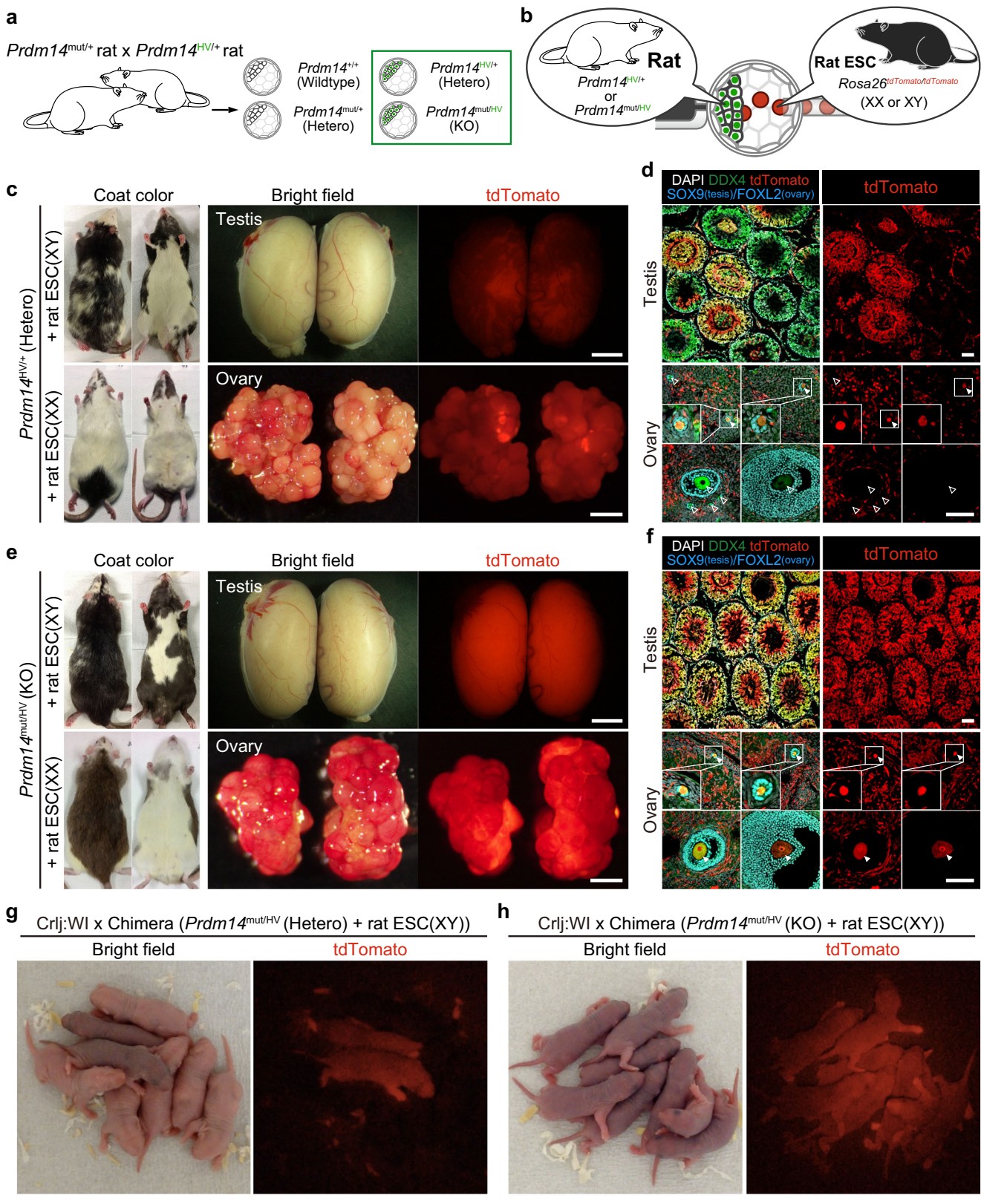

kidney, the chimeras grew into normal adults without any impairment, suggesting host-derived cells developmentally rescued the defect. Indeed, the chimeras exhibited a lack of contribution of *Pax2*/*Pax8* KO cells to renal lineages, including the ureters (Fig. 2b, Supplementary Fig. 2b–d). In contrast, all the germline in testes and ovaries of *Prdm14* KO rats are composed of tdTomato expressing cells (Fig. 2b), confirming

our previous result (Fig. 1), that the germline is complemented by donor *Pax2*/*Pax8* KO rat ESCs. Next, we crossed male and female chimeras with *Pax2*/*Pax8* KO gametes to obtain *Pax2*/*Pax8* double KO rats. As expected, all the offspring lacked kidneys and ureters (Fig. 2c, Table 1). Thus, we could efficiently generate double KO rats using blastocyst complementation through the germline.

**Fig. 1 Blastocyst complementation in the germline using *Prdm14* KO rat blastocysts and allogenic rat ESCs. a** Collection of blastocysts after crossing *Prdm14*mut/+ rats with *Prdm14*HV/+ rats. Prdm14-H2BVenus positive blastocysts were selectively used for subsequent blastocyst injection. **b** Schematics showing injection of male (XY) or female (XX) *Rosa26*tdTomato/tdTomato rat ESCs into Prdm14-H2BVenus positive blastocysts. **c** Chimerism of coat color, testis, and ovary in *Prdm14*HV/+ chimeric rats with *Rosa26*tdTomato/tdTomato rat ESCs. Scale bar in testis: 5 mm, in the ovary: 1 mm. **d** Immunohistochemical staining of testis and ovary of chimeras in Fig.1c. Some (filled arrowheads) but not all DDX4 positive germ cells originate from tdTomato positive donor ESCs. Scale bars: 100 μm. **e** Chimerism of coat color, testis, and ovary in *Prdm14*mut/HV chimeric rats with *Rosa26*tdTomato/tdTomato rat ESCs. Scale bar in testis: 5 mm, in the ovary: 1 mm. **f** Immunohistochemical staining of testis and ovary of chimeras in Fig.1e. All the DDX4 positive germ cells originate from tdTomato positive donor ESCs. Scale bars: 100 μm. **g** Offspring obtained by crossing *Prdm14*HV/+ chimeric rats with wild-type Crlj: WI rats. Ubiquitous tdTomato expression shows successful germline transmission of injected *Rosa26*tdTomato/tdTomato rat ESCs. **h**. Offspring obtained by crossing *Prdm14*mut/HV chimeric rats with wild-type Crlj: WI rats. All the neonates ubiquitously expressed tdTomato indicating perfect germline transmission of *Rosa26*tdTomato/tdTomato rat ESCs.

**Table 1 Blastocyst injection and germline transmission of ESCs.**

| Donor PSC | | | *Prdm14* genotype in host embryo | No. (%) of injected embryos | | No. (%) of pups | | No. of live sex-matched chimeras | | |
|---|---|---|---|---|---|---|---|---|---|---|
| Cell type | Genotype | Gender | | Transferred | Full-term | Analyzed | Chimeras | Analyzed | KO | GT |
| Rat ESC | *Rosa26*tdTomato/tdTomato | XY | KO or Hetero | 65 | 33 (51) | 31 | 24 (77) | 14 | 3 | 2 |
| Rat ESC | *Rosa26*tdTomato/tdTomato | XX | KO or Hetero | 78 | 30 (38) | 30 | 15 (50) | 9 | 4 | 3 |
| Rat ESC | *Pax2/Pax8* KO | XY | KO or Hetero | 173 | 69 (40) | 67 | 33 (49) | 17 | 9 | 9 |
| Rat ESC | *Pax2/Pax8* KO | XX | KO or Hetero | 254 | 126 (50) | 126 | 37 (29) | 15 | 12 | 5[b] |
| Rat ESC | MAC | XY | WT or Hetero[a] | 218 | 132 (61) | 132 | 112 (85) | 53 | – | 0 |
| Rat ESC | MAC | XY | KO or Hetero | 22 | 18 (82) | 18 | 16 (89) | 7 | 5 | 5 |
| Mouse ESC | *Blimp1*Tg(BAC-mEGFP)/+/ CAG-nmScarlet | XY | KO or Hetero | 32 | 17 (53) | 17 | 9 (53) | 6 | 3 | 2 |
| Mouse iPSC | CAG-EGFP | XY | KO or Hetero | 44 | 17 (39) | 17 | 9 (53) | 6 | 2 | 2 |

[a]Wildtype or Prdm14-H2BVenus negative (*Prdm14*mut/+ or *Prdm14*+/+) blastocysts were used as the control experiments.
[b]We tested 5 out of 12 *Prdm14*HV/mut (KO) chimeras for the germline transmission.

To further demonstrate the potential of blastocyst complementation via the germline and compare with available methods to produce genetically modified animals, we next used a rat ESC line carrying a MAC. MAC is an exogenous chromosome capable of replicating and segregating as an independent chromosome[27]. MAC is a useful tool for creating humanized model rodents that recapitulate human disorders. MACs inserts are several megabase long sequences, which cannot be achieved by conventional embryonic transgenesis approaches[28]. Technically, the introduction of MAC requires cell-based transgenesis using PSCs or male germline stem cells[27,29]. Thus, the successful production of the MAC carrying animals relies mainly on the quality of the cells. Here, we used male rat ESCs carrying a MAC containing EGFP expression cassette (Fig. 2a, d). When we injected the MAC rat ESCs into wild-type or *Prdm14* heterozygous rat blastocysts, we were unable to obtain any chimeras showing signs of germline contribution of the MAC rat ESCs in the testes, despite testing over 50 sex-matched chimeras (Table 1). While most of them showed EGFP fluorescence in the testis (Fig. 2e), only the somatic cells surrounding the germ cells were EGFP-positive in the sections (Fig. 2f), indicating that the MAC rat ESC line we used in this study shows almost no germline contribution. In contrast, in chimeras obtained from *Prdm14* KO rat blastocysts, we could observe seminiferous tubules filled with EGFP-positive cells in the adult testes (Fig. 2e). The tubules contained male germ cells at all stages of development, including mature spermatozoa (Fig. 2f). To obtain the next generation carrying the MAC, we collected and sorted EGFP-positive round spermatids from the chimeric testes (Fig. 2g). Since expression of *Prdm14-H2BVenus* is absent in adult testes of both *Prdm14*HV/+ and *Prdm14*mut/HV[30], we can selectively sort donor MAC rat ESC-derived cells from the adult testis. Next, we performed a round spermatid injection (ROSI) into oocytes from wild-type female rats. After embryo transfer (ET), we successfully obtained 21 pups, and 8 of them showed

ubiquitous expression of EGFP (Fig. 2h, Table 2). All the adults show the black eye and coat color, corresponding to the original ESCs (Supplementary Figure 1d). Notably, EGFP-positive adult shows a presence of the MAC in all the cells in the brain (Supplementary Fig. 1c), suggesting that round spermatid cells generated using *Prdm14* KO blastocysts carry the MAC to the next generation.

**Generation of functional mouse spermatids in rat via interspecific blastocyst complementation in the germline**. Next, we investigated whether blastocyst complementation in the germline applies to an interspecific setting between the mouse and rat. To monitor specification and subsequent development of mouse PGCs in rat, we used mouse ESCs established from the *Blimp1-membrane (m) EGFP* reporter transgenic mouse (*Blimp1*Tg(BAC-mEGFP)/+), a faithful line to monitor PGCs until gonadal stage[31]. The EGFP protein in *Blimp1* expressing mouse PGCs localizes at the membrane. Whereas, the Venus signal in Prdm14-H2BVenus rat PGCs is nuclear. Therefore the mouse PGCs are readily distinguishable from rat PGCs. In addition, for visualizing the contribution to somatic cells as well as the germline after blastocyst injection, we introduced the CAG promoter-driven *mScarlet-3xNLS (nmScarlet)* expressing vector via a PiggyBac system into mouse ESCs (Supplementary Figure 3a). Upon injection of the labeled mouse ESC line into mouse blastocysts, we confirmed their competence towards forming chimeras and in germline transmission (Supplementary Fig. 3b–h). We injected the mouse ESCs into Prdm14-H2BVenus positive rat blastocysts and confirmed contribution to the germline during development (Fig. 3a). At 6 days after ET corresponding to embryonic day (E) 9–9.5 when rat PGCs specify, *Prdm14* KO PGCs expressing *H2BVenus* were detected in the posterior epiblast (dashed line in Fig. 3b), although the rat PGCs show reduced proliferation and fail to activate key PGC markers due to the lack of

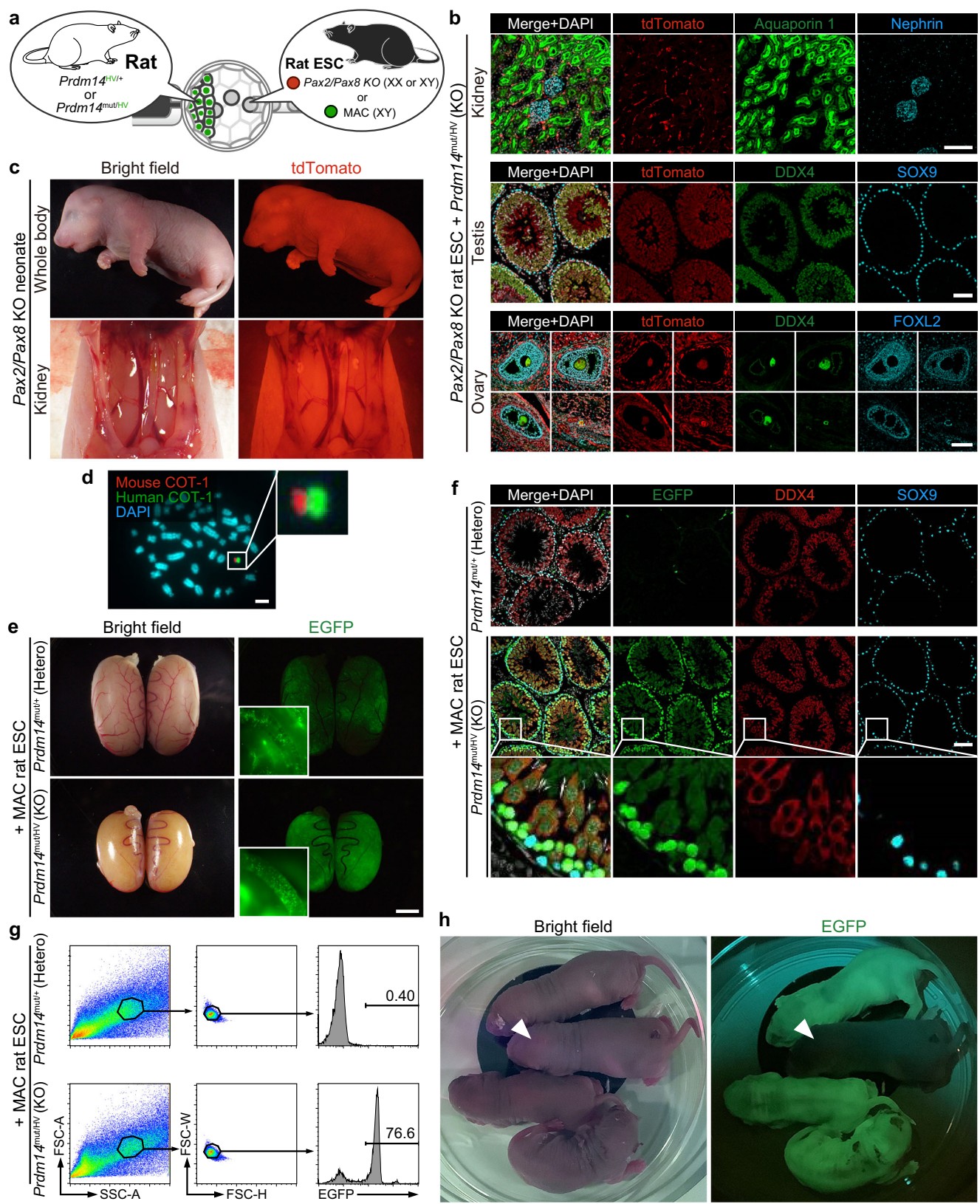

*Prdm14*[32]. Strikingly however, Blimp1-mEGFP/CAG-nmScarlet positive cells were also observed in the PGC cluster, suggesting that the rat environment is conducive for mouse ESC-derived PGC specification (Fig. 3b). At 12 days post ET (E15-15.5), mouse PGCs were undetectable in *Prdm14* heterozygous rat gonads, although nmScarlet positive somatic cells surrounding rat PGCs were present

(Supplementary Fig. 4b). In contrast, Blimp1-mEGFP/CAG-nmScarlet positive mouse PGCs expressing DAZL, a key gonadal PGC marker, colonized *Prdm14* KO rat gonads normally devoid of rat PGCs (Fig. 3c, Supplementary Fig. 4b). The somatic contribution of mouse ESCs in rats is, however, not as high as in mice (Supplementary Fig. 4c). In the adult chimeras with *Prdm14* KO,

**Fig. 2 Generation of genetically modified rats using blastocyst complementation in the germline. a** Schematics showing injection of genetically modified rat ESCs into Prdm14-H2BVenus positive blastocysts. **b** Immunohistochemical staining of adult kidney, testis, and ovary in *Prdm14*mut/HV chimeric rats with *Pax2/Pax8* KO rat ESCs (*Rosa26*tdTomato/tdTomato background). While no contribution of the rat ESCs was observed in Aquaporin-1 positive proximal tubule and Nephrin positive glomeruli in the kidney, all germ cells both in the testis and ovary originate from rat ESCs. A male ESC line and a female ESC line were used for the testis and ovary, respectively. Both ESC lines dominantly contribute to DDX4 positive germ cells, and partially to the surrounding somatic cells (SOX9 positive Sertoli cells or FOXL2 positive granulosa cells, respectively). Scale bars: 100 μm. **c** A representative picture of neonates obtained by crossing male and female *Prdm14*mut/HV chimeric rats with *Pax2/Pax8* KO rat ESC-derived gametes. The neonate exhibits a ubiquitous expression of tdTomato and a complete lack of kidney. **d** FISH image of rat ESC carrying mouse artificial chromosome (MAC). Scale bars: 5 μm. **e** Testis of MAC rat ESC chimeras. While both shows EGFP fluorescence, only the *Prdm14*mut/HV chimeric rat testis contains a seminiferous tubule filled with EGFP-positive MAC rat ESC-derived germ cells. Representative seminiferous tubes of each chimera are shown in the insets. Scale bar: 5 mm. **f** Immunohistochemical staining of the testis from chimera generated by injection of MAC rat ESCs into Prdm14-H2BVenus positive blastocysts. Insets show higher magnification of basal compartments of the testis in *Prdm14*mut/HV chimeric rats. Scale bar: 100 μm. **g** FACS patterns of testicular cells in *Prdm14*mut/HV or *Prdm14*mut/+ chimeric rats with MAC rat ESCs. **h** Offspring generated by round spermatid injection (ROSI) of FACS-sorted MAC rat ESC-derived round spermatids into wild-type rat oocytes. An arrowhead shows EGFP negative sibling.

**Table 2 Injection of ESC-derived round spermatids into oocytes.**

| Donor spermatid | | Host oocyte | No. (%) of oocytes | | | | No. (%) of pups | |
|---|---|---|---|---|---|---|---|---|
| Origin | Genotype | | Injected | Survived | Cleaved | Transferred | Born | Fluorescence |
| Rat ESC | MAC | Slc:SD (Rat) | 208 | 149 (72) | 15 (10) | 143 | 21 (15) | 8 (38) |
| Mouse ESC | *Blimp1*Tg(BAC-mEGFP)/+/*CAG-nmScarlet* | Slc:BDF1 (Mouse) | 87 | 72 (83) | 38 (53) | 68 | 20 (29) | 20 (100) |
| Mouse iPSC | *CAG-EGFP* | Slc:BDF1 (Mouse) | 173 | 169 (98) | 153 (91) | 162 | 37 (23) | 16 (43) |

two out of three testes showed nmScarlet expression within the seminiferous tubules, and strikingly, normal spermatogenesis could be observed in the section (Fig. 3d, e, Table 1). In the epididymis, we could also observe mature sperm with mScarlet signal in the nucleus (Fig. 3f), but they somehow rarely show motility after being harvested into human tubal fluid (HTF) medium suitable for mouse sperm[33]. Similarly, hamster sperm also showed abnormalities when transplanted into mouse testes[34], suggesting that spermiogenesis might be impaired in xenogeneic environments; the precise reasons merits further considerations. To test the function of mouse ESC-derived male germ cells generated in rats, we performed ROSI of the FACS-sorted mouse ESC-derived round spermatid into oocytes collected from a wild-type mouse. We successfully obtained healthy mouse offspring, and all of them showed ubiquitous nmScarlet expression, and almost half of them inherited *Blimp1-mEGFP* transgene at a Mendelian ratio (7/20 [35.0%], Fig. 3g, Table 2)

Finally, we also tested mouse induced pluripotent stem cells (iPSCs) from *CAG-EGFP* transgenic mouse[6,35,36] as donor cells for blastocyst complementation in the germline (Table 1). As with mouse ESCs, EGFP-positive mouse iPSC-derived spermatogenesis was observed in *Prdm14*mut/HV rat testes (Fig.3h, i). Notably, the haploid cells were able to produce viable mice after injection into mouse oocytes (Fig. 3j). Almost half of the mice expressed EGFP ubiquitously (16/37 [43%], Fig. 3j, Table 2), and all the pups checked here inherited retroviral integrations of a reprograming factor used for the establishment of iPSCs (Fig. 3k). Taken together, we demonstrated that interspecific blastocyst complementation in the germline can yield functional mouse spermatids in rats.

## Discussion
We demonstrate applications of blastocyst complementation in the germline using *Prdm14* KO rat embryos as hosts, with allogenic rat ESCs as well as xenogenic mouse ESCs as donor cells. Since our approach allows efficient germline transmission, we were able to produce rat lines bearing elaborate, as well as precise genetic modifications. Lacking competition from host germ cells

unlike in conventional chimeras, our method enriches for germ cells with desired genotypes.

By using the blastocyst complementation method, we demonstrated the efficient generation of *Pax2/Pax8* double mutant rats lacking both renal parenchyma and ureters. For blastocyst complementation in the kidney, we previously used blastocysts from *Sall1* KO mice or rats[7,10]. However, since *Sall1* KO animals show defects in the renal mesenchyme but not in the ureteric bud during renal development[10,37], the ureters in the chimera consist of both injected donor PSCs and host-derived cells. Such contamination of host-derived tissues might prevent efficient engraftment of PSC-derived kidneys in transplantation therapy. Thus, for complete complementation of organs, host lines with multiple mutations will prove beneficial in the future. Blastocyst complementation in the germline is an effective way to generate such elaborate genetically modified animals.

Successful germline transmission is a stringent criterion for determining PSC pluripotency[30]. While mechanisms behind the success or failure of germline transmission are not fully known, the genetic background of the donor or host cells, chromosomal abnormalities, or PSC sources such as those from in vitro reprogramming are known to influence the germline potential[38,39]. In our study, a rat ESC line carrying a MAC contributed normally to somatic tissues but not to the germline in wild-type or heterozygous animals where the host germ cells are present. This is likely due to the clone we used since previous reports have demonstrated successful germline transmission of PSCs with similar artificial chromosome vectors[27,28]. Even when retaining a normal karyotype, not all rat PSC lines show germline transmission; indeed, line-to-line and clone-to-clone variations are common[21,40]. Specifically, the complete lack of the host germ cells in *Prdm14* KO rat however allowed rescue of the MAC rat ESCs, which formed functional gametes, demonstrating that the MAC rat ESC line itself can develop as germ cells. Yet, in the presence of host germ cells, the MAC bearing ESC-exhibit a disadvantage since germ cells from them are competitively eliminated from the germline. Therefore, the use of germline-deficient animals, might also test the competency of PSCs for

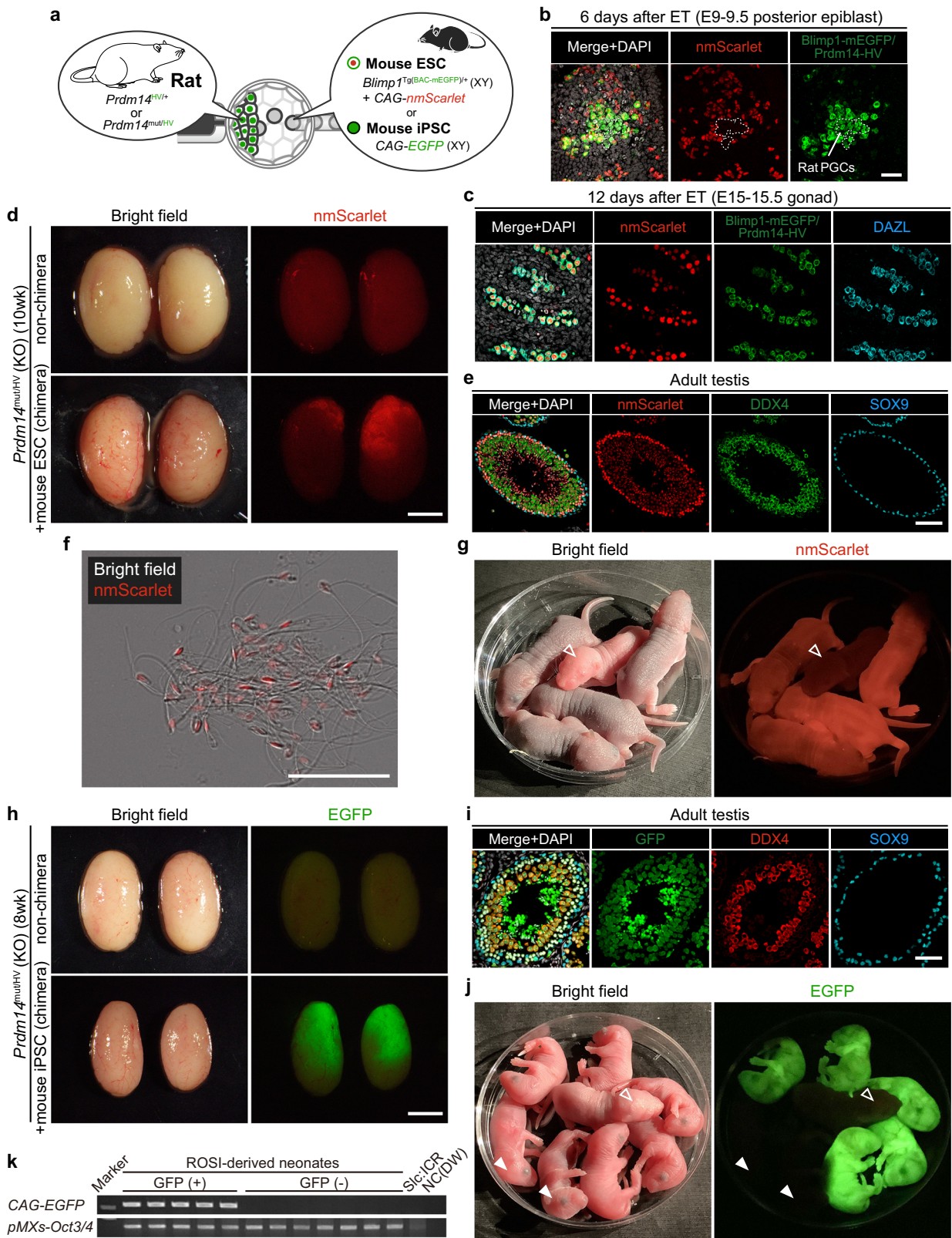

germline transmission in the absence of competition from host germ cells.

Interspecific blastocyst complementation using *Prdm14* KO rat with mouse ESCs or iPSCs can generate functional mouse male germ cells in rats. While rat PSCs can contribute to germline even in wild-type mouse[12,41,42], we hardly find mouse PGCs in rat gonads in the presence of the host rat PGCs (Supplementary

Fig. 4b). Interestingly, at the onset of PGC specification and prior to their migration toward gonads, a significant number of mouse PGCs clustered with host rat PGCs in most of the *Prdm14*[HV/+] rat chimeras (Supplementary Fig. 4a). Thus, these chimeras with a high contribution of xenogenic cells might be eliminated due to the abnormality often seen in the interspecific chimeras[6,36], or due to competition between mouse and rat PGCs, particularly in

**Fig. 3 Interspecific blastocyst complementation in germline generates functional mouse spermatids in rats. a** Schematics showing injection of mouse ESCs or iPSCs into Prdm14-H2BVenus positive rat blastocysts. **b** Wholemount immunostaining of the posterior part of $Prdm14^{mut/HV}$ chimeric embryo with mouse ESCs at 6 days after embryo transfer (ET: embryonic day (E) 9–9.5). Dashed line shows cluster composed of H2BVenus expressing $Prdm14^{mut/HV}$ rat PGCs and mEGFP/nmScarlet double-positive mouse PGCs. Scale bar: 50 μm. **c** Immunohistochemical staining of male gonads of $Prdm14^{mut/HV}$ chimeric fetus at 12 days after ET (E15-15.5). mEGFP/nmScarlet/DAZL triple-positive mouse PGCs colonized in rat gonad. In contrast, no rat PGCs are present due to defect in the Prdm14 gene. Scale bar: 50 μm. **d** Images of adult testis. The upper testis is a control testis from $Prdm14^{mut/HV}$ rat, and the lower one is from $Prdm14^{mut/HV}$ chimeric rats with mouse ESCs. Scale bar: 5 mm. **e** Immunohistochemical staining of representative seminiferous tubules. The tubules filled with nmScarlet positive cells show normal spermatogenesis originate from mouse ESC-derived germ cells. Scale bar: 50 μm. **f** Mouse ESC-derived sperm collected from the epididymis of $Prdm14^{mut/HV}$ chimeric rats. Scale bar: 50 μm. **g** Postnatal day 3 neonates generated by ROSI of mouse ESC-derived round spermatids into wild-type mouse oocytes. An open arrowhead shows control Slc:ICR neonate. **h** Images of the adult testis. The upper testis is a control testis from $Prdm14^{mut/HV}$ rat, and the lower one is from $Prdm14^{mut/HV}$ chimeric rats with mouse iPSCs. Scale bar: 5 mm. **i** Immunohistochemical staining of representative seminiferous tubules. The tubules filled with EGFP-positive cells show normal spermatogenesis originate from mouse iPSC-derived germ cells. Scale bar: 50 μm. **j** Postnatal day 1 neonates generated by ROSI of mouse iPSC-derived round spermatids into wild-type mouse oocytes. An open arrowhead shows control Slc:ICR neonate, and arrowheads show EGFP negative siblings. **k** A representative genotyping result to detect the CAG-EGFP transgene and retroviral integration of a reprograming factor, pMXs-Oct3/4, used for the establishment of iPSCs.

the gonad where they proliferate actively. In any case, mouse PSCs do not readily contribute to the germline in rats compared with rat PSCs in the mouse. Previously, we demonstrated that the contribution of xenogenic PSCs in interspecific chimeras between mouse and rat varies greatly from organ to organ[36]. For instance, in the kidney, rat PSCs contribute poorly to the metanephric mesenchyme, the origin of renal parenchyma during embryonic development, leading to failure in the generation of rat kidney in Sall1 KO mouse[7]. Conversely, mouse kidneys are generated in the Sall1 KO rat owing to the efficient contribution of mouse PSCs to the metanephric mesenchyme[10]. Thus, the combination of the species in chimera creation dictates the success of contribution to the desired tissue or organ. Contribution to tissues such as the neuron or the germline also requires careful consideration due to ethical concerns when blastocyst complementation is applied to human organ regeneration in livestock[43]. It is likely that such contribution might be low or negligible, depending on the particular species used as host.

Interspecific chimeras between the mouse and rat are generally smaller than intra-specific chimeras[6,36]. Thus, the greater evolutionary distance between the donor and host species may limit the extent of chimerism, which will affect the contribution to the germline. However, the use of recipient embryos incapable of generating a host germline, such as in Prdm14 KO rats, increases the likelihood of xenogenic germ cells even with PSCs that contribute poorly to somatic tissues. Since in vitro reprogramming technology enables PSCs derivation from endangered species[44,45], the generation of xenogenic germ cells in germline-deficient animals could provide a useful approach to generate functional gametes for the preservation of such species.

Taken together, blastocyst complementation in the germline shown is a robust system for efficient generation of desired animals and xenogenic germ cells from PSCs.

## Methods

**Animals.** Crlj:WI (RGD ID: 2312504) rats were purchased from Charles River Laboratories Japan, Inc. (Kanagawa, Japan). Slc:SD (RGD ID: 12910483) rats, C57BL/6NCrSlc, B6D2F1/Slc, and Slc:ICR mice were purchased from SLC Japan (Shizuoka, Japan). All experiments were performed in accordance with the animal care and use committee guidelines of the National Institute for Physiological Sciences.

**Culture of PSC lines.** Undifferentiated rat ESCs (WDB-Rosa26em1(RT2)Nips-ES2/Nips, WDB/Nips-ES1/Nips, RGD ID: 10054010) were maintained on mitomycin-C-treated mouse embryonic fibroblasts in N2B27 medium (all components were purchased from Thermo Fisher Scientific, MA) containing 1 μM PD0325901 (Axon, Groeningen, The Netherlands), 3 μM CHIR99021 (Axon), and 1000 U/ml of rat leukemia inhibitory factor (Millipore, Bedford, MA). A female $Rosa26^{tdTomato/tdTomato}$ rat ESC line used in Figs. 1 and 2 was established from

blastocysts obtained from an intercross of $Rosa26^{tdTomato/tdTomato}$ rats (WDB-$ROSA26^{em1(RT2)Nips}$) according to published protocol[46].

For making Pax2/Pax8 double KO rat ESCs, pX330-U6-Chimeric_BB-CBh-hSpCas9 (Addgene plasmid # 42230) vectors expressing Cas9 protein with gRNA target for coding sequence of Pax2 and Pax8 were introduced by reverse transfection method using lipofectamine 2000 (Thermo Fisher Scientific) as described before[47]. After transfection, we carried out limiting dilution to enrich single cell-derived colonies and screened for mutants by PCR and subsequent Sanger sequencing using the following primer sets: Pax2 (Fw, 5′- CCTCCATTT TTCAGCTCTGC-3′, and Rv, 5′-CCCTCACTCCAAGTCTCAGC-3′), Pax8 (Fw, 5′-TCCCCAAATGGTGTGTTTCT-3′, and Rv, 5′- CGTTGGTGTTCCTT GTCCTT-3′).

For making rat ESCs carrying MAC, MAC was transferred to rat ESCs via microcell-mediated chromosome transfer[28]. Briefly, rat ESCs (BLK2i-1, RGD ID: 10054010) were fused with microcells prepared from the donor hybrid cells, and selected with G418 (150 μg/mL). CHO cells containing a MAC with human chromosome 21 region, which was developed previously[48], was used as donor hybrid cells in this study.

Undifferentiated mouse ESCs (established from $Blimp1^{Tg(BAC-mEGFP)/+}$ mouse in previous reports[49,50]) and mouse iPSCs (GT3.2 line[6]) were maintained on 0.1% gelatin-coated plates in Ndiff227 medium (Takara Bio, Shiga, Japan) containing 1 μM PD0325901, 3 μM CHIR99021, 1,000 U/ml of mouse leukemia inhibitory factor (Millipore), and 5% fetal calf serum (FCS, batch tested for mouse ESCs; Thermo Fisher Scientific). To monitor the ESC-derived cells in chimeras, a PiggyBac vector containing CAG-driven mScarlet with 3x nuclear localization signals cassettes were transfected into mouse ESCs with PBase plasmid by reverse transfection method. The transfected mouse ESCs were seeded on gelatin-coated plates and 48 h later, 0.8 μg/ml puromycin (Sigma-Aldrich, MO) was added to the culture medium for selection. Puromycin-resistant and nmScarlet positive mouse ESC colonies were picked and used for the experiment.

**Genotyping.** To detect transgenes in germline transmitted pups or adults, genomic DNA was analyzed by PCR using the following primer sets: tdTomato (Fw, 5′-GCG AGGAGGTCATCAAAGAG-3′, and Rv, 5′-GATGACGGCCATGTTGTTGT-3′), CAG-nmScarlet (Fw, 5′-GCCTCTGCTAACCATGTTCA-3′, and Rv, 5′-CTTGTAC AGCTCGTCCATGCCG-3′), Blimp1-mEGFP (Fw, 5′-TAAACGGCCACAAGTT CAGCG-3′, and Rv, 5′-CTCCTTGAAGTCGATGCCCTT-3′), CAG-EGFP (Fw, 5′-GCCTCTGCTAACCATGTTCA-3′, and Rv, 5′-CTCCTTGAAGTCGATG CCCTT-3′), pMXs-Oct3/4 (Fw, 5′-CCCTGGGGATGCTGTGAGCCAAGG-3′, and Rv, 5′-CCCTTTTTCTGGAGACTAAATAAA-3′).

**FISH analyses.** The trypsinized rat ESCs and homogenized brain samples were incubated for 15 min in 0.075 M KCl, fixed with methanol and acetic acid (3:1), and then slides were prepared using standard methods. FISH analyses were performed using fixed metaphase spreads of each cell hybrid using digoxigenin-labeled (Roche, Basel, Switzerland) DNA [mouse COT-1 DNA] and biotin-labeled DNA [human COT-1 DNA (Invitrogen)][51]. Chromosomal DNA was counterstained with DAPI (Sigma-Aldrich). Images were captured using an AxioImagerZ2 fluorescence microscope (Carl Zeiss GmbH).

**Embryo manipulation.** A step-by-step protocol describing blastocyst complementation can be found at our protocol published previously[52].

For rat blastocyst injection, blastocysts were collected in mR1ECM medium from the oviduct and uterus of rats at 4.5 days post coitum (dpc). For micromanipulation, rat or mouse ESCs were trypsinized and suspended in the ESC medium, respectively. A piezo-driven micro-manipulator (Prime Tech, Tokyo, Japan) was used to drill into the zona pellucida and trophectoderm under the microscope and 7–8 rat ESCs were introduced into blastocyst cavities near the

inner cell mass. After the injection, the blastocysts were transferred into the uterus of pseudopregnant recipient Crlj:WI rats (3.5 dpc).

For mouse blastocyst injection, morula stage embryos were collected in KSOM medium from oviduct and uterus of mice at 2.5 dpc, and then cultured for 1 day until they reached the blastocyst stage. Mouse ESCs were trypsinized and suspended in the ESC medium. As in the rat blastocyst injection, 7–8 mouse ESCs were introduced into blastocyst cavities near the inner cell mass. After injection, the blastocysts were transferred into the uterus of pseudopregnant recipient Slc:ICR mice (2.5 dpc).

For rat ROSI, seminiferous tubules were dissociated mechanically and EGFP-positive MAC rat ESC-derived rat round spermatids were collected with the SH800 cell sorter (SONY) according to a published protocol[53]. Sorted round spermatids were suspended in PBS containing glucose, sodium lactate, and FCS and used as donor cells. Metaphase II stage oocytes were collected in HEPES-KRB medium from superovulated Slc:SD rats and denuded the cumulus with 0.1% hyaluronidase treatment. Rat oocytes were then pretreated with 5 mM ionomycin for 5 min and incubated for 40 min in mKRB medium until injection. After ROSI, the fertilized embryos were treated for 4 h with 5 μg/mL cycloheximide (Sigma-Aldrich) in mKRB. After follow-up incubation in mKRB for 1 day, embryos were transferred into the oviduct of pseudo-pregnant Crlj:WI rats.

For mouse ROSI, nmScarlet positive mouse ESC-derived mouse round spermatids were collected with the SH800 cell sorter and used as donor cells. Metaphase II stage oocytes were collected in CZB medium from B6D2F1/Slc mice and denuded the cumulus with 0.1% hyaluronidase treatment. Before ROSI, mouse oocytes were treated with 10 mM $SrCl_2$ for 20 min. After follow-up incubation of the mouse embryos in CZB for 1 day, embryos were transferred into the oviduct of pseudo-pregnant Slc:ICR mice.

**Histological analysis**. Tissues were fixed in 4% PFA for 4 h to O/N at RT depending on the size. Whole embryos and embryonic gonads were fixed in 4% PFA for 30 min at RT or for 6 h at 4 °C. For cryosections, the fixed embryos were embedded in OCT compound. Each sample was incubated with primary antibodies for 1–2 h at RT or O/N at RT or 4 °C, and then with fluorescent-conjugated secondary antibodies with DAPI (FUJIFILM Wako, Osaka, Japan) for 1 h at RT or O/N at 4 °C. Antibodies used here are listed in Supplementary Table 2. The embryos and cryosections were observed under confocal laser scanning microscopy (FV3000; Olympus, Tokyo, Japan). For imaging, we carefully performed all the experiments with appropriate controls. To judge positive fluorescent signals, we obtained the images alongside samples of chimera or non-chimera that have partial or no contribution of donor cells.

**Progeny testing to confirm germline transmission of $Rosa26^{tdTomato/tdTomato}$ rat ESCs**. Chimeric rats generated by injection of male or female $Rosa26^{tdTomato/tdTomato}$ rat ESCs were crossed with Crlj:WI (RGD ID: 2312504) rats. Since ESCs are homozygous for Rosa26-tdTomato, all the pups with germline transmission ubiquitously express tdTomato fluorescence in all the tissues. In addition, since rat ESCs originate from WDB/Nips black rat (RGD ID: 7411634), the resulting pups from germline transmission of the rat ESCs also clearly show black coat color and black eye by the effect of the dominant gene. Thus, we judged the number of pups resulting from the germline transmission of tdTomato fluorescence, and the coat and eye color appearance.

**Blood chimerism analysis**. Peripheral blood samples obtained from each chimera were hemolyzed by red blood cell lysis buffer. After washing the cells with PBS containing 3% FCS, the percentage of donor PSC-derived cells were determined by fluorescent signals in the blood cells detected by SH800 cell sorter.

**Data reproducibility**. IF images in Figs. 1d, f, 2b, f, 3b, c, e, i, Supplementary Figs. 2b,d, 3b,c, 4a,b were representatives of multiple sections at different projections (6<). Karyotyping images in Fig. 2d and Supplementary Fig. 1c were representatives obtained from multiple cells on each slide. Image in Fig. 3f is representative of two animals. Genotyping in Fig. 3k, Supplementary Figs. 1a and 3h was performed one time with appropriate loading controls.

**Reporting summary**. Further information on research design is available in the Nature Research Reporting Summary linked to this article.

## Data availability

The authors declare that all data supporting the findings of this study are available within the article and its supplementary information files or from the corresponding author upon reasonable request. Source data are provided with this paper.

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

## Acknowledgements

We thank members of the Hirabayashi lab, in particular, Keiko Yamauchi and Megumi Hashimoto for help with animals, and Minako Ohnishi for secretarial support. We also thank Dr. Roopsha Sengupta for editing and providing critical input to the manuscript. This work was supported by grants from Grant-in-Aid for Scientific Research from the Japan Society for the Promotion of Science (18H02367) to M.H. and T.K., (18H05548) and (18H05544) to T.K., JST CREST (JPMJCR18S4) to Y.K., and LEAP-AMED (JP18gm0010002) to H.N. and M.H., AMED (JP18bm0704022) to T.K., and The Great Britain Sasakawa Foundation, Butterfield Awards in Medicine and Health (B115) to T.K. and M.A.S.

## Author contributions

T.K., T.G., M.O., and M.H. designed the study. T.K. wrote the manuscript with help of M.A.S. T.K., M.O., N.N., M.S., and M.H. performed embryo manipulation. T.K., T.G., M.O., and M.H. analyzed the chimeras. T.K., T.G., F.Y., and R.T. performed molecular biological analysis and cell culture. N.K., K.K., and Y.K. performed chromosome transfer and FISH analysis. M.A.S. provided materials. H.N. and S.H. supervised the project.

## Competing interests

The authors declare no competing interests.
