## [Peer Review File · Nature Communications]

Reviewers' Comments:

Reviewer #1:

Remarks to the Author:

In this manuscript, the authors attempt to generate a more efficient method for germline transmission of pluripotent stem cells using blastocyst complementation technology.

Unfortunately, the need for these studies is poorly articulated. Additionally, these studies lack novelty, proper validation, and are somewhat disjointed. The specific concerns are offered below.

1. The authors suggest that PSCs allow for cell based genetic modifications that gene editing technologies do not. This is a vague statement that requires significant clarification as it serves as the center point of the presumed hypothesis. Moreover, the hypothesis is not clear and, as such, it is difficult to discern whether the hypothesis has been adequately tested.

2. The authors indicate that germline transmission of genetic modifications is variable and inefficient, presumably due to competition between host and donor. In this manuscript, the authors generate a niche by deleting *Prdm14* which was demonstrated by others (in 2008) to impair development of germ line cells. The mutant host niche (or WT host) is repopulated using a variety of modified donor cells (rat or mouse). The results support the conclusion that the generation of a niche (when compared with no niche) benefits the survival of donor cells. This has been demonstrated numerous times in the literature, particularly in rodents (and by this group as well). The results do not, however, support the conclusion that germline transmission is less variable or more efficient using this approach (presumably, the goal of this study). Specifically, chimerism is only quantified by evaluating fluorescence in whole animals or whole mount organs (fluorescent or not fluorescent). This 'quantification', the methods for which are not described, is presented in Table 1. The data in Table 1 are confusing and do not seem to coincide with the text based descriptions. Moreover, this type of whole mount, dark field imaging is easily confounded and should be carefully described and controlled and complemented with other techniques. What were the excitation parameters, how was imaging performed, what cameras/filters were used, were identical parameters used for every litter? How was the presence or absence of fluorescence validated? Currently, there is no description of this in the methods. The authors marked both the host cells (H2BVenus) and donor cells (tdTomato). Chimerism could be quantified based on the expression of these markers yet only representative immunohistochemical images were presented as qualitative data. Moreover, it was observed that tdTomato cells contribute to somatic cells in both WT and mutant hosts yet the authors claim that the donor cells exclusively contribute to germline. There appears to be no basis for this conclusion. Moreover, since only whole animals or the gonads have been evaluated, it is not possible to understand how donor cells have contributed to other lineages. This is important information that will address the feasibility of this technique. Instead, it was only indicated that other organ systems developed normally.

3. One of the most significant ethical concerns with interspecies chimerism is the presence of donor cells in the host germline or brain. Exogenous cell population in the germline, has the potential to generate unintended mutations in offspring. One commonly proposed solution to this issue is to neuter the host to prevent germline transmission. This approach would work against the stated goals of the current study. How have the authors accounted for unintended germline mutations and transmissions in this study? How does the risk of unintended mutations outweigh any benefits of the use of this technique?

4. The authors claim to have generated a novel double mutant rat line (*Pax2/Pax 8* double mutant; lacks kidneys and ureters). This mouse line was already generated by this group as an improvement over the *Sall1* mutant (which lacks kidney but has ureters). What is the purpose of this mutant rat line? This seems to represent an incremental advance over the development of the double knockout. Moreover, how is this approach superior to the use of multiplex gene editing?

5. Fig S1. There is no validation of the *Pax2/Pax8* frameshift mutation.

6. Fig 2. The authors use a green fluorescent marker to indicate donor cells (MAC cells; cytoplasmic marker) from host (*Prdm14*mut/HV; also green fluorescent but a nuclear marker). They use FACS to sort these two markers but provide no evidence for the validation of this approach.

7. Figure 2d, why is human COT-1 present?

Reviewer #2:

Remarks to the Author:

This manuscript from Kobayashi and addresses a major barrier in use of rat gene-targeted genetic models – extremely low rate of germline transmission. The authors' approach, using Prdm14 KO rat ESCs which are incompetent for germline specification and complementation with gene-targeted Prdm14-wt ESCs, is both elegant and effective. The authors provide beautiful and convincing data demonstrating 1) high-efficiency germline transmission (in both testes and ovaries) of Rosa-TdTomato genotype from rat ESCs into Prdm14KO (and at lower rate in Prdm14Het) blastocysts, 2) germline transmission of gene-targeted rat ESCs (Pax2/Pax8 double KO) leading to nephric agenesis, 3) germline transmission of rat ESCs carrying a mouse artificial chromosome, but only in Prdm14-KO blasts, and 4) generation of functional mouse spermatids from mouse ESCs injected into Prdm14-KO rat blasts (although this requires use of ROSI, rather than natural mating, to propagate). I found these studies to be largely very well executed, properly controlled and appropriately interpreted. Although blastocyst injections are not technically trivial, the ability to produce germline gene targeted rats enabled by this study would be transformative to the biomedical research enterprise because rats are a more suitable model for a number of diseases. The authors, however, are cautioned to not make claims about the likely generalizability of this approach to other species (abstract lines 46-47, discussion lines 303-312) for three reasons. First, while the authors demonstrate xenogeneic mouse spermatogenesis in rat testes, the reciprocal produced by rat spermatogonial stem cell transplantation into mouse testes has been known for many years (also hamster to mouse), but it is also known that higher mammals SSCs are incompetent for spermatogenesis in mouse testes. Thus, it seems very likely the same will be true for rat as a xenogeneic germ cell host. Second, if this approach is used without germline incompetent rat blastocysts for other species, comparable germline incompetent blasts (e.g., Prdm1/14, or Nanos2/3 KOs) would first need to be produced in those species. Third (and least concerning), a PSC source would need to be produced in the species of interest. I raise this issue because it is premature to even speculate that this approach is widely generalizable, especially in the abstract, and should be more cautiously introduced. Indeed, I felt this point detracted from the otherwise very strong manuscript.

The following specific criticisms should also be addressed:

1. Figure 1c/e – these images, while very nice, do not sufficiently resolve cells to demonstrate to the reader the extent of donor germline contribution. This could be addressed by histology retaining the tdtomato label. I realize that the pups in Fig. 1d/f demonstrate this outcome in a different way, but additional evidence is required to support the statement on line 122 that ALL germ cells in chimeras were TdT+.
2. Fig. 2f & Lines 263-266 – it is very curious why germline transmission of MAC-containing rat ESCs was never observed with Prdm14Het blastocyst – the authors should provide some explanation for why this is.
3. Fig. 3g – It is not able that mouse sperm in rat epididymides were immotile – what is the explanation for this result?

Reviewer #3:

Remarks to the Author:

This is an interesting paper with novel findings. The authors have used blastocyst complementation chimeras with rat host embryos mutant in Prdm14, a gene essential for primordial germ cell development, and donor pluripotent stem cells (PSC) of four types: (i) Rat ESCs homozygous for a ubiquitously expressing tdTomato gene, where all germ cells in the chimeras expressed the reporter, as did all offspring. (ii) Rat PSC null mutant for both Pax2 and Pax8, which leads to the complete absence of the kidney. This allowed the efficient creation of

anephric rats without a complex breeding program. (iii) Rat PSC carrying a mouse artificial chromosome (MAC), which failed to contribute to the germline in normal rat chimeras colonised the gonads when Prdm14 mutant blastocysts were used. (iv) Mouse PSC gave rise to immotile sperm, but to round spermatids that gave live born animals after ROSI. All situations provide good examples of how the system might be used generally, with the last being of potential importance for rescuing endangered species. The work is well done, however several aspects were a little minimal.

1. Relying largely on one fluorescent marker, as with experiment (i) and (iii) is not ideal. While the simplest and most likely interpretation of the data is the one given, it is possible to have cell fusion and transfer of the marker gene into host cells. Because the ES cells and host embryos were from different strains, it would have been straightforward to genotype resulting animals to ensure that they came from the former. With the Pax2/Pax8 double mutants, (experiment (ii)) it is very likely that these must have come from the ES cells carrying these mutated genes.

2. Experiment (iv) is interesting, but it is unclear why the mouse sperm were immotile. The mouse ESCs used were expressing mEGFP from the Blimp gene and had been further genetically altered to carry CAG-nmScarlet via a PiggyBAC method. While these were apparently competent to undergo germline transmission, we are only shown 3 pups in Fig. S2 and are not given any idea of the efficiency of this. This data should be provided. Moreover, the levels of nmScarlet varied substantially between the 3 pups. Was the line used to make chimeras clonal in origin ?

3. It is also possible that the degree of manipulation has caused a problem in spermiogenesis, which is just tolerated in the mouse, but not in the rat testis. This could include high levels of expression of fluorescent proteins. Given that it is not necessary to have any marker if the only sperm that could develop are mouse, the experiment should be repeated using unmanipulated mouse ESCs. This would also better reflect any application of the methods for rescuing endangered species.

We thank you for all your time and effort in reviewing our manuscript **NCOMMS-20-20379** entitled "Blastocyst complementation for robust germline transmission and generation of functional mouse spermatids in rats". We present a revised version where we have addressed all the concerns, and have incorporated valuable suggestions by the reviewers. This has strengthened and improved the manuscript. Here is a point-wise response to the reviewer's comments.

General Comment

ESCs and iPSC are widely used for genetic manipulations, for studies in *in vitro* models, and for generating transgenic colonies from them, notably of mice and rats. The latter are crucial for studies on gene functions *in vivo* that inform normal development with implications for human diseases. Efficient transmission of manipulated ESCs and iPSCs through the germline is an essential procedure for such studies.

Germline transmission of PSCs, and particularly iPSCs can vary and has at times proved difficult if not impossible. At the very least, an efficient procedure that reduces the numbers of experimental animals is by itself highly desirable. Here we also explored a possibility of interspecies germline transmission to determine if such a procedure could in principle be developed for animal breeding, and perhaps contribute to the development of future methodologies for the rescue of endangered mammals when iPSCs are the only cells available from them.

Our full response to the reviewers' comments are as follows:

Reviewer #1 (Remarks to the Author):

In this manuscript, the authors attempt to generate a more efficient method for germline transmission of pluripotent stem cells using blastocyst complementation technology. Unfortunately, the need for these studies is poorly articulated. Additionally, these studies lack novelty, proper validation, and are somewhat disjointed. The specific concerns are offered below.

We provide a clear rationale for our study, as also pointed in our **General comment** above, and included in the Introduction of the revised manuscript.

1. The authors suggest that PSCs allow for cell based genetic modifications that gene

editing technologies do not. This is a vague statement that requires significant clarification as it serves as the center point of the presumed hypothesis. Moreover, the hypothesis is not clear

and, as such, it is difficult to discern whether the hypothesis has been adequately tested.

The initial studies on gene functions can sometimes start with modified PSCs using *in vitro* models. The same cells can then be used to generate transgenic animals through germline transmission for studies under physiological conditions *in vivo*. PSCs are also essential in multiple scenarios, for example in the introduction of a mammalian chromosome vector that are structurally similar to native chromosome ¹. As far as we know, gene editing technologies cannot substitute for such elaborate modifications. In some cases, PSC might be a preferred route while in other cases gene editing approach might be preferred. We do not imply that PSC approach is better, which depends on the purpose of investigation.

Rat is a widely used model organism for studying physiology, behavior, toxicology and various pathologies such as neurodegenerative diseases. However, low germline transmission rate in rats has been the biggest barrier for the use of genetically modified rat. Here, we set out to test if this barrier can be overcome through blastocyst complementation in a germ cell-less niche. We provide various examples of rat mutant lines created using this approach. This method expands the utility and potential of the rat model system, and provides an outlook for the use of xenogeneic chimera.

Although direct gene editing in mammalian zygotes is a powerful tool to obtain mutant animals, precise modifications such as introduction of point mutations or genes of interest via gene targeting is not efficient yet ². In addition, manipulation of zygotes often causes mosaicism in the founder generation ³. To segregate the desired phenotype requires considerable effort and time. Thus, the use of well-characterized PSCs after gene editing is an advantage in terms of the reliability and reproducibility. We have clarified these points in the revised manuscript (Page 4 line 57-62).

2. The authors indicate that germline transmission of genetic modifications is variable and inefficient, presumably due to competition between host and donor. In this manuscript, the authors generate a niche by deleting Prdm14 which was demonstrated by others (in 2008) to impair development of germ line cells. The mutant host niche (or WT host) is repopulated using a variety of modified donor cells (rat or mouse). The results support the conclusion that the generation of a niche (when compared with no niche) benefits the survival of donor cells. This has been demonstrated numerous times in the literature, particularly in rodents (and by this group as well). The results do not, however, support the conclusion that

germline transmission is less variable or more efficient using this approach (presumably, the goal of this study). Specifically, chimerism is only quantified by evaluating fluorescence in whole animals or whole mount organs (fluorescent or not fluorescent). This 'quantification', the methods for which are not described, is presented in Table 1. The data in Table 1 are confusing and do not seem to coincide with the text based descriptions. Moreover, this type of whole mount, dark field imaging is easily confounded and should be carefully described and controlled and complemented with other techniques. What were the excitation parameters, how was imaging performed, what cameras/filters were used, were identical parameters used for every litter? How was the presence or absence of fluorescence validated? Currently, there is no description of this in the methods. The authors marked both the host cells (H2BVenus) and donor cells (tdTomato). Chimerism could be quantified based on the expression of these markers yet only representative immunohistochemical images were presented as qualitative data. Moreover, it was observed that tdTomato cells contribute to somatic cells in both WT and mutant hosts yet the authors claim that the donor cells exclusively contribute to germline. There appears to be no basis for this conclusion. Moreover, since only whole animals or the gonads have been evaluated, it is not possible to understand how donor cells have contributed to other lineages. This is important information that will address the feasibility of this technique. Instead, it was only indicated that other organ systems developed normally.

To address the reviewer's concerns, we have added a new table (**Table S1**) showing quantification for germline transmission efficiency and chimerism. In **Table S1**, we have listed individual chimeras analyzed in the experiment of **Fig. 1**, and the number of germline transmission after mating each chimera with wildtype animals. The efficiency of germline transmission matches with contribution of donor PSCs to functional gametes. As an example for a somatic lineage other than the germline in the same table, we have also listed individual chimerism in peripheral blood (PB) measured by flow cytometry. While only 0–35% of pups from *Prdm14*^{Venus/+} (Hetero) chimeric rats showed germline transmission despite high chimerism in PB of the chimeras, all pups (100%) from *Prdm14*^{Venus/mut} (KO) chimeric rats showed germline transmission, indicating that PSCs did contribute exclusively to germline in the absence of host germ cells.

For imaging, we have carefully performed all the experiments with appropriate controls. We imaged chimera alongside non-chimera samples under the same settings, whereby the positive and negative fluorescent signals can be clearly distinguished. We have added these details in the methods section (Page 31 line 635-638).

In addition to the imaging of fluorescent signal, we have also confirmed germline

transmission by genotyping as well as coat/eye color, which is different between the host (albino, red) and the donor (black, black) (**Fig. 1g,h, Fig. S1b,d**). We believe that presenting multiple parameters strengthens evidence for germline transmission of donor PSCs.

3. One of the most significant ethical concerns with interspecies chimerism is the presence of donor cells in the host germline or brain. Exogenous cell population in the germline, has the potential to generate unintended mutations in offspring. One commonly proposed solution to this issue is to neuter the host to prevent germline transmission. This approach would work against the stated goals of the current study. How have the authors accounted for unintended germline mutations and transmissions in this study? How does the risk of unintended mutations outweigh any benefits of the use of this technique?

We acknowledge the ethical concerns raised by the reviewer in the context of interspecific blastocyst complementation for organ regeneration. However, the aim of our study is the use of donor PSCs for germline transmission and to generate germ cells. Thus, donor cell contribution to tissues other than the germline is not of consequence in this system. To prevent the risk of unintended mutations, PSCs can be extensively screened before blastocyst complementation in the germline, which is among the advantages of using PSCs in this approach. On the other hand, efficient germline transmission of PSCs would be beneficial, by reducing the numbers of animals used for generating rodent models. The approach might also be useful for animal breeding, and perhaps for the rescue of endangered mammals.

4. The authors claim to have generated a novel double mutant rat line (Pax2/Pax 8 double mutant; lacks kidneys and ureters). This mouse line was already generated by this group as an improvement over the Sall1 mutant (which lacks kidney but has ureters). What is the purpose of this mutant rat line? This seems to represent an incremental advance over the development of the double knockout. Moreover, how is this approach superior to the use of multiplex gene editing?

The purpose behind creating *Pax2/Pax8* rats was to show that blastocyst complementation in the germline is a viable approach for creating double mutants with germline transmission, without a complex breeding program requiring large numbers of animals and the costs involved in doing so.

In a previous report we generated *Pax2/Pax8* mouse ES cell lines to prevent renal lineage formation, which is different to the approach shown here⁴. The aim of generating *Pax2/8 KO*

rat line is, as the reviewer pointed out, an improvement over the *Sall1* mutant for both kidney and ureter via blastocyst complementation, which will be used for future study. Multiplex gene editing often causes not only targeted double mutant phenotype but also various genotypes such as heterozygous of either one or two genes. Mosaicism after gene editing in zygote also causes complexity of the analysis. Thus, we believe that our approach using well-characterized PSCs is reliable for creating multiple mutants.

5. Fig S1. There is no validation of the Pax2/Pax8 frameshift mutation.

To validate the *Pax2/Pax8* frameshift mutation, we analyzed E11-11.5 chimeric fetus generated by injection of *Pax2/Pax8* KO rat ESCs into blastocysts (**Fig. S2c**). Immunostaining revealed that tdTomato+ cells in metanephric mesenchyme (MM) adjacent to E-cadherin+ nephric duct (ND) did not show expression of both *Pax2* and *Pax8* likely due to the mutation (**Fig. S2d**). Furthermore, *Pax2/Pax8* KO cells did not contribute to ND at all, which is akin to failure of ND formation in *Pax2/Pax8* mutant mice⁵ (**Fig. S2d**).

6. Fig 2. The authors use a green fluorescent marker to indicate donor cells (MAC cells; cytoplasmic marker) from host (Prdm14mut/HV; also green fluorescent but a nuclear marker). They use FACS to sort these two markers but provide no evidence for the validation of this approach.

Expression of *Prdm14-H2BVenus* is undetectable in the germline as well as the other tissues for at least 4 weeks and afterwards both in male and female⁶. Furthermore, there are no *Prdm14-H2BVenus* positive germ cells in *Prdm14^{venus/mut}* (KO). Thus, all the GFP positive cells come from MAC rat ESCs. Indeed, all the rats generated from GFP positive spermatids have black coat color with black eyes (**Fig. 2h, Fig. S1d**) suggesting the origin is donor PSC. To prevent confusion, we have added a sentence in the revised manuscript (Page 10 line 191-193).

7. Figure 2d, why is human COT-1 present?

As shown in the methods section (*Culture of pluripotent stem cell lines*), we used a MAC with human chromosome fragments, which explains the presence of human COT-1.

Reviewer #2 (Remarks to the Author):

This manuscript from Kobayashi and addresses a major barrier in use of rat gene-targeted genetic models – extremely low rate of germline transmission. The authors' approach, using Prdm14 KO rat ESCs which are incompetent for germline specification and complementation with gene-targeted Prdm14-wt ESCs, is both elegant and effective. The authors provide beautiful and convincing data demonstrating 1) high-efficiency germline transmission (in both testes and ovaries) of Rosa-TdTomato genotype from rat ESCs into Prdm14KO (and at lower rate in Prdm14Het) blastocysts, 2) germline transmission of gene-targeted rat ESCs (Pax2/Pax8 double KO) leading to nephric agenesis, 3) germline transmission of rat ESCs carrying a mouse artificial chromosome, but only in Prdm14-KO blasts, and 4) generation of functional mouse spermatids from mouse ESCs injected into Prdm14-KO rat blasts (although this requires use of ROSI, rather than natural mating, to propagate). I found these studies to be largely very well executed, properly controlled and appropriately interpreted. Although blastocyst injections are not technically trivial, the ability to produce germline gene targeted rats enabled by this study would be transformative to the biomedical research enterprise because rats are a more suitable model for a number of diseases.

We thank the reviewer for the encouraging, supportive and constructive comments.

The authors, however, are cautioned to not make claims about the likely generalizability of this approach to other species (abstract lines 46-47, discussion lines 303-312) for three reasons. First, while the authors demonstrate xenogeneic mouse spermatogenesis in rat testes, the reciprocal produced by rat spermatogonial stem cell transplantation into mouse testes has been known for many years (also hamster to mouse), but it is also known that higher mammals SSCs are incompetent for spermatogenesis in mouse testes. Thus, it seems very likely the same will be true for rat as a xenogeneic germ cell host. Second, if this approach is used without germline incompetent rat blastocysts for other species, comparable germline incompetent blasts (e.g., Prdm1/14, or Nanos2/3 KOs) would first need to be produced in those species. Third (and least concerning), a PSC source would need to be produced in the species of interest. I raise this issue because it is premature to even speculate that this approach is widely generalizable, especially in the abstract, and should be more cautiously introduced. Indeed, I felt this point detracted from the otherwise very strong manuscript.

We agree with the reviewer, while we believe that our technique has future potential, many hurdles have to be overcome for practical application beyond rodents. In line with the reviewer's suggestion, we have modified the abstract and the manuscript to tone down the generalizability of the application (Page 3 line 46-47, Page 16 line 332- Page 17 line 335).

The following specific criticisms should also be addressed:

1. *Figure 1c/e – these images, while very nice, do not sufficiently resolve cells to demonstrate to the reader the extent of donor germline contribution. This could be addressed by histology retaining the tdtomato label. I realize that the pups in Fig. 1d/f demonstrate this outcome in a different way, but additional evidence is required to support the statement on line 122 that ALL germ cells in chimeras were TdT+.*

We have added histology data of chimeric testis and ovary in revised **Fig. 1d, f** supporting the statement. In addition, in the revised **Table S1**, we listed the efficiency of germline transmission of individual chimeras. Together, the data support that all the germ cells originate from donor ESCs in chimera with *Prdm14^{Venus/mut}* (KO) hosts.

2. *Fig. 2f & Lines 263-266 – it is very curious why germline transmission of MAC-containing rat ESCs was never observed with Prdm14Het blastocyst – the authors should provide some explanation for why this is.*

We speculate this has to do with the clone we used, since previous reports demonstrated MAC/HAC containing mouse/rat ESCs showing successful germline transmission^{7, 8}. We and the other group previously showed not all rat PSCs contribute to germline and there are obvious line-to-line and clone-to-clone variations^{9, 10}. We have added this statement in **Page 15 line 290-294**.

3. *Fig. 3g – It is not able that mouse sperm in rat epididymides were immotile – what is the explanation for this result?*

The reason is not clear yet, but even with mouse iPSCs, we found that some but not all mouse sperm in rat epididymis show motility. A previous study demonstrated abnormalities in hamster sperm after interspecific testicular cell transplantation in the mouse epididymis¹¹. Thus, xenogeneic environments might cause some impairment during spermiogenesis, albeit the function of haploid cell is normal as we have shown. We have added this explanation in **Page 12 line 240-243**.

--

Reviewer #3 (Remarks to the Author):

This is an interesting paper with novel findings. The authors have used blastocyst complementation chimeras with rat host embryos mutant in Prdm14, a gene essential for primordial germ cell development, and donor pluripotent stem cells (PSC) of four types: (i) Rat ESCs homozygous for a ubiquitously expressing tdTomato gene, where all germ cells in the chimeras expressed the reporter, as did all offspring. (ii) Rat PSC null mutant for both Pax2 and Pax8, which leads to the complete absence of the kidney. This allowed the efficient creation of anephric rats without a complex breeding program. (iii) Rat PSC carrying a mouse artificial chromosome (MAC), which failed to contribute to the germline in normal rat chimeras colonised the gonads when Prdm14 mutant blastocysts were used. (iv) Mouse PSC gave rise to immotile sperm, but to round spermatids that gave live born animals after ROSI. All situations provide good examples of how the system might be used generally, with the last being of potential importance for rescuing endangered species. The work is well done, however several aspects were a little minimal.

We thank the reviewer for the encouraging, supportive and constructive comments.

1. Relying largely on one fluorescent marker, as with experiment (i) and (iii) is not ideal. While the simplest and most likely interpretation of the data is the one given, it is possible to have cell fusion and transfer of the marker gene into host cells. Because the ES cells and host embryos were from different strains, it would have been straightforward to genotype resulting animals to ensure that they came from the former. With the Pax2/Pax8 double mutants, (experiment (ii)) it is very likely that these must have come from the ES cells carrying these mutated genes.

We have added representative genotyping results to detect a transgene for experiment (i) and FISH showing presence of MAC in the brain for experiment (iii) (**Fig. S1a, c**). Furthermore, since both coat and eye colors are different between ESCs (black, black) and host embryos (Albino, red), we have added clear pictures of F1 pups after mating with wild type animals in **Fig. 1g,h, Fig. S1b,d**. We believe these additional data further support germline transmission of donor ESCs.

2. Experiment (iv) is interesting, but it is unclear why the mouse sperm were immotile. The mouse ESCs used were expressing mEGFP from the Blimp gene and had been further genetically altered to carry CAG-nmScarlet via a PiggyBAC method. While these were apparently competent to undergo germline transmission, we are only shown 3 pups in Fig. S2 and are not given any idea of the efficiency of this. This data should be provided. Moreover, the levels of nmScarlet varied substantially between the 3 pups. Was the line used to make chimeras clonal in origin ?

The reason for sperm immotility is not clear yet. A previous study demonstrated abnormalities in hamster sperm after interspecific testicular cell transplantation into mouse epididymis ¹¹. Thus, xenogeneic environments might cause some impairment during spermiogenesis, albeit functionally the haploid cells are normal, as we show here. We have added this explanation in **Page 12 line 240-243**.

In addition to 3 pups expressing nmScarlet, we have added an image of the adults showing dark agouti coat color (ES cell origin; **Fig. S3f**). In addition, genotyping data show presence of *CAG-nmScarlet* in all the adults and *Blimp1-mEGFP* transgenes in some (**Fig. S3h**). We have also confirmed that peripheral blood from all the adults showed nmScarlet expression. Notably, expression patterns of nmScarlet varied likely due to random insertion of the transgene via the PiggyBac system (**Fig. S3g**). We have briefly added the explanation in the figure legend.

3. It is also possible that the degree of manipulation has caused a problem in spermiogenesis, which is just tolerated in the mouse, but not in the rat testis. This could include high levels of expression of fluorescent proteins. Given that it is not necessary to have any marker if the only sperm that could develop are mouse, the experiment should be repeated using unmanipulated mouse ESCs. This would also better reflect any application of the methods for rescuing endangered species.

To answer the reviewer's comments in a timely fashion, instead of unmanipulated mouse ESCs, here we show results for mouse iPSCs derived from *CAG-EGFP* transgenic mouse, since we had already set them up to generate chimeras during submission. The original transgenic mouse line and the iPSC line was well characterized previously ^{4, 12, 13}. In the revised **Fig. 3h-k**, we successfully demonstrated generation of functional mouse iPSC-derived spermatids in chimera with *Prdm14^{venus/mut}* rats, which is similar to the result of mouse ESCs. Interestingly, even the use of the mouse iPSCs, some but not all the mouse sperm in rat epididymis show motility, suggesting again this might be due to xenogenic

environment. Further extensive study will be required to reveal the causes, but we believe this is out of the current focus. We have added the result in **Page 12 line 244 – Page 13 line 253**.

While, the cell line we showed here is different from the reviewer's suggestion, we believe the use of iPSCs is more practical than the use of ESCs for future applications related to rescuing endangered species, and possibly for breeding of domestic animals.

References

1. Doherty AMO, Fisher EMC. Microcell-mediated chromosome transfer (MMCT): small cells with huge potential. *Mammalian Genome* **14**, 583-592 (2003).
2. Oji A, *et al.* CRISPR/Cas9 mediated genome editing in ES cells and its application for chimeric analysis in mice. *Scientific Reports* **6**, 31666 (2016).
3. Mehravar M, Shirazi A, Nazari M, Banan M. Mosaicism in CRISPR/Cas9-mediated genome editing. *Dev Biol* **445**, 156-162 (2019).
4. Yamaguchi T, *et al.* An interspecies barrier to tetraploid complementation and chimera formation. *Sci Rep* **8**, 15289 (2018).
5. Bouchard M, Souabni A, Mandler M, Neubuser A, Busslinger M. Nephric lineage specification by Pax2 and Pax8. *Genes Dev* **16**, 2958-2970 (2002).
6. Kobayashi T, *et al.* Germline development in rat revealed by visualization and deletion of Prdm14. *Development* **147**, (2020).
7. Kazuki Y, *et al.* Humanized UGT2 and CYP3A transchromosomal rats for improved prediction of human drug metabolism. *Proc Natl Acad Sci U S A* **116**, 3072-3081 (2019).
8. Kazuki K, *et al.* Highly stable maintenance of a mouse artificial chromosome in human cells and mice. *Biochem Biophys Res Commun* **442**, 44-50 (2013).
9. Hirabayashi M, *et al.* A retrospective analysis of germline competence in rat embryonic stem cell lines. *Transgenic research* **22**, 411-416 (2013).

10. Blair K, Leitch HG, Mansfield W, Dumeau CE, Humphreys P, Smith AG. Culture parameters for stable expansion, genetic modification and germline transmission of rat pluripotent stem cells. *Biol Open* **1**, 58-65 (2012).
11. Ogawa T, Dobrinski I, Avarbock MR, Brinster RL. Xenogeneic Spermatogenesis Following Transplantation of Hamster Germ Cells to Mouse Testes¹. *Biology of Reproduction* **60**, 515-521 (1999).
12. Kobayashi T, *et al.* Generation of rat pancreas in mouse by interspecific blastocyst injection of pluripotent stem cells. *Cell* **142**, 787-799 (2010).
13. Okabe M, Ikawa M, Kominami K, Nakanishi T, Nishimune Y. 'Green mice' as a source of ubiquitous green cells. *FEBS Lett* **407**, 313-319 (1997).

Reviewers' Comments:

Reviewer #1:

Remarks to the Author:

In re-reviewing this manuscript, it does seem that there is merit in this approach though enthusiasm is still reduced as the knock out models utilized here were previously described. In other words, the phenotypes are not unique but it is acknowledged that the combination of these two phenotypes is new.

The authors have provided some level of quantification of their results but more description is required. In Table S1, the authors now indicate the number of pups that were analyzed and the number that demonstrate chimerism. The method of analysis, however, is not provided. Specifically, how did the authors arrive at 100% TdTomato+ germ line cells in the Prdm14 KO groups? Was this based on positive fluorescence of the whole animal? If so, this is an inadequate level of rigor. At very least, the authors should provide the # of animals analyzed, the # of sections analyzed and the # of cells analyzed in the germ line. Given that the conclusions of this study rely on one single method (+/- fluorescence), this level of quantification is the bare minimum that is required for this study. Also, the legend for this table lacks adequate description. What is Blood Chimerism (note misspelling) and how was it measured?

The authors have yet to adequately describe the methods and controls that were utilized to perform immunohistochemistry and imaging. Simply stating that they were carefully evaluated with proper controls is inadequate. What controls were employed for imaging and for immunohistochemistry? How were the antibodies characterized? Please refer to my initial review for all details that are required.

There remains no data to support the statement that 'the injected rat ESCs dominantly and perhaps exclusively contributed towards the germline'. Based on the data provided in Table S1, this statement is false and, therefore, should be modified to remove any reference to exclusive contribution to the germline.

Because cells do not exclusively track to the germline, and because the authors boldly state that this technology could rescue endangered species, the onus is on the authors to discuss how this will be done safely and without a significant mutation of a species. The authors rebut that they will carefully screen PSCs before blastocyst complementation but how will that be performed and for what will they screen to ensure that mutations to a species do not occur? This is an important societal question and one that must be addressed if the authors are to propose this technology for improved germline transmission of transgenic rats and for the rescue of endangered species. The major reason that genetically modified rats were difficult to generate historically was due to the lack of rat ESCs. That hurdle no longer exists and, therefore, the authors should acknowledge this fact within the manuscript.

Regarding the green fluorescent markers to separate donor cells from host, the authors indicate that there are no H2BVenus positive germ cells in the KO and therefore there is no potential for confounding signals. This statement presumes, however, that the KO was successful. If the KO were not successful would all the green fluorescent be attributed to MAC?

Reviewer #2:

Remarks to the Author:

The authors of this revised manuscript from Kobayashi and colleagues have done a very nice job of addressing my prior concerns and comments. I have no further critiques to offer. This is a very nice manuscript that makes a valuable contribution to the transgenesis field by leveraging the male germline.

Here is a point-wise response to the reviewer's comments.

Reviewer #1 (Remarks to the Author):

In re-reviewing this manuscript, it does seem that there is merit in this approach though enthusiasm is still reduced as the knock out models utilized here were previously described. In other words, the phenotypes are not unique but it is acknowledged that the combination of these two phenotypes is new.

We are very grateful to the reviewer for granting that the combination of the two phenotypes is new.

The authors have provided some level of quantification of their results but more description is required. In Table S1, the authors now indicate the number of pups that were analyzed and the number that demonstrate chimerism. The method of analysis, however, is not provided. Specifically, how did the authors arrive at 100% TdTomato+ germ line cells in the Prdm14 KO groups? Was this based on positive fluorescence of the whole animal? If so, this is an inadequate level of rigor. At very least, the authors should provide the # of animals analyzed, the # of sections analyzed and the # of cells analyzed in the germ line. Given that the conclusions of this study rely on one single method (+/- fluorescence), this level of quantification is the bare minimum that is required for this study. Also, the legend for this table lacks adequate description. What is Blood Chimerism (note misspelling) and how was it measured?

We judged germline transmission of the ESCs by tdTomato fluorescent and black eye/coat color. We also confirmed that tdTomato fluorescent positive pups have *tdTomato* transgene inserted into *Rosa26* locus of injected ESCs (**Fig.S1a**). Notably, since we used *Rosa26^{tdTomato/tdtomato}* homozygous rat ESCs as we described in the **Methods**, all the pups come from ESC-derived germ cells express tdTomato in all tissues under the control of endogenous *Rosa26* promoter. Besides, the ESCs originate from rats with black coat color, WDB/Nips (RGD ID: 7411634). Accordingly, all the offspring that inherited the genotype have black eye/coat color after crossing with CrIj:WI (RGD ID: 2312504) due to the dominant gene. Our experimental design and results constitute definitive proof of germ line transmission of donor ESCs, which never occurs in animals in the absence of germline transmission. We have included further explanation on how we tested the germline

transmission in the **Methods** (Page31 Line631 - Page32 Line 641).

The number of chimeric animals we analyzed for the germline transmission is indicated in **Table 1** and **Table S1**. For individual animals, please see ID (far left column) in Table S1. We tested 3 males and 4 females in *Prdm14* KO chimera, and 5 males and 6 females in heterozygous for germline transmission of ESCs by crossing with wildtype animals.

For blood chimerism, peripheral blood obtained from each chimera are hemolyed by red blood cell lysis buffer, and the percentage of tdTomato positive cells in the blood cells were analyzed by FACS. We have added the detail in the **Methods** (Page32 Line643- 647).

The authors have yet to adequately describe the methods and controls that were utilized to perform immunohistochemistry and imaging. Simply stating that they were carefully evaluated with proper controls is inadequate. What controls were employed for imaging and for immunohistochemistry? How were the antibodies characterized? Please refer to my initial review for all details that are required.

For immunohistochemistry, as we mentioned in the **Method**, we took images of heterozygous and KO chimera at the same time. The contrast of the negative/positive signal for tdTomato or EGFP fluorescence is distinct in the section of heterozygous chimeric rats. As controls for **Fig.3c**, we attach the non-chimera control taken simultaneously with the indicated samples (**Rebuttal Fig.1a**). For **Fig.3e,f**, we have added lower magnification images that contain testicular tubes without fluorescent signal due to lack of donor mouse PSC derived germ cells (**Rebuttal Fig.1b,c**), which is an appropriate stained control.

The antibodies we used in this study are listed in **Supplementary Table S2**, and all of them were appropriately validated either by the distributors or by ourselves. We have added a data showing controls without primary antibody staining for GFP and RFP signal in **Rebuttal Fig.1d**.

There remains no data to support the statement that 'the injected rat ESCs dominantly and perhaps exclusively contributed towards the germline'. Based on the data provided in Table S1, this statement is false and, therefore, should be modified to remove any reference to exclusive contribution to the germline.

In response to the reviewer's comment, we have removed the indicated sentence.

Because cells do not exclusively track to the germline, and because the authors boldly state that this technology could rescue endangered species, the onus is on the authors to discuss how this will be done safely and without a significant mutation of a species. The authors rebut that they will carefully screen PSCs before blastocyst complementation but how will that be performed and for what will they screen to ensure that mutations to a species do not occur? This is an important societal question and one that must be addressed if the authors are to propose this technology for improved germline transmission of transgenic rats and for the rescue of endangered species.

The careful screening of PSCs we suggested in the rebuttal letter has already been conducted for the application of PSCs for regenerative medicine. Researchers screen the iPSC lines by multiple analysis as a preclinical quality control ¹, and importantly, the qualified PSC-derivatives have already been used to cure the patients in clinical trial ². New technologies are emerging for whole genome sequencing and amplification (WGA), which are accurate and reproducible even at the resolution of single cells (<https://bioskryb.com>). Informed decisions will be possible on screening PSCs for germline transmission when strict quality control is necessary for applying our technology, which is feasible and practically applicable to any species.

The major reason that genetically modified rats were difficult to generate historically was due to the lack of rat ESCs. That hurdle no longer exists and, therefore, the authors should acknowledge this fact within the manuscript.

We have added the sentence in the **introduction** (Page5 Line 90-93).

Regarding the green fluorescent markers to separate donor cells from host, the authors indicate that there are no H2BVenus positive germ cells in the KO and therefore there is no potential for confounding signals. This statement presumes, however, that the KO was successful. If the KO were not successful would all the green fluorescent be attributed to MAC?

Even if they are not KO, i.e. heterozygous, expression of *Prdm14-H2BVenus* is absent in adult testes since neither germ cells nor somatic cells express *Prdm14* later than postnatal day 8 (P8), and then afterwards ³. We have added the images with appropriate wildtype controls in **Rebuttal Fig.1e**. Notably, also in mice, expression of *Prdm14* is restricted in the embryonic germline development ⁴. Thus, as the reviewer acknowledged, all the GFP signal in the adult rat come from MAC. We have added the references to clarify it on **Page10, Line189-191**.

Rebuttal Figure 1

- a.** Immunohistochemical staining of male gonads of *Prdm14^{HV/+}* (Hetero) non-chimeric fetus at 12 days after ET (E15-15.5). Non-specific signals for nmScarlet (RFP antibody) were not observed in the section. This image was taken simultaneously with images in **Fig.3c** and **Fig.S4b**. Scale bar: 50 μ m.
- b.** Immunohistochemical staining of adult testis of chimera generated by injection of nmScarlet positive mouse ESCs into *Prdm14^{HV/+}* (KO) blastocyst. The tubules without fluorescent signal due to lack of donor mouse PSC derived germ cells is an appropriate stained control. Scale bar: 100 μ m.
- c.** Immunohistochemical staining of adult testis of chimera generated by injection of EGFP positive mouse iPSCs into *Prdm14^{HV/+}* (KO) blastocyst. Scale bar: 100 μ m.
- d.** Validation of antibodies against GFP and RFP (anti-mCherry antibody in **Supplemental Table 2**) using section of chimeric gonad generated by injection of mouse ESCs into *Prdm14^{HV/+}* (Hetero) rat. Staining with these antibodies specifically enhance appropriate signals compared with control without primary antibodies. Scale bar: 100 μ m.
- e.** Immunostaining of *Prdm14^{HV/+}* (Hetero) and *Prdm14^{+/+}* (Wildtype) gonads at indicated stages, which is cited from our previous report³. Scale bar: 100 μ m.

References

1. Doi D, *et al.* Pre-clinical study of induced pluripotent stem cell-derived dopaminergic progenitor cells for Parkinson's disease. *Nature Communications* **11**, 3369 (2020).
2. Mandai M, *et al.* Autologous Induced Stem-Cell-Derived Retinal Cells for Macular Degeneration. *The New England journal of medicine* **376**, 1038-1046 (2017).
3. Kobayashi T, *et al.* Germline development in rat revealed by visualization and deletion of Prdm14. *Development* **147**, (2020).
4. Yamaji M, *et al.* Critical function of Prdm14 for the establishment of the germ cell lineage in mice. *Nat Genet* **40**, 1016-1022 (2008).

Reviewers' Comments:

Reviewer #1:

Remarks to the Author:

The authors have adequately responded to my queries.